# Towards Understanding Variants of Invariant Risk Minimization through the Lens of Calibration

**Yoshida Kotaro**[1*]**, Hiroki Naganuma**[2,3*]

*yoshida.k.bl@m.titech.ac.jp, naganuma.hiroki@mila.quebec,*
[1] *Tokyo Institute of Technology,* [2] *Mila - Quebec AI Institute,* [3] *Université de Montréal,*

**Reviewed on OpenReview:** *https://openreview.net/forum?id=9YqacugDER*

## Abstract

Machine learning models traditionally assume that training and test data are independently and identically distributed. However, in real-world applications, the test distribution often differs from training. This problem, known as out-of-distribution (OOD) generalization, challenges conventional models. Invariant Risk Minimization (IRM) emerges as a solution that aims to identify invariant features across different environments to enhance OOD robustness. However, IRM's complexity, particularly its bi-level optimization, has led to the development of various approximate methods. Our study investigates these approximate IRM techniques, using the consistency and variance of calibration across environments as metrics to measure the invariance aimed for by IRM. Calibration, which measures the reliability of model prediction, serves as an indicator of whether models effectively capture environment-invariant features by showing how uniformly over-confident the model remains across varied environments. Through a comparative analysis of datasets with distributional shifts, we observe that Information Bottleneck-based IRM achieves consistent calibration across different environments. This observation suggests that information compression techniques, such as Information Bottleneck are potentially effective in achieving model invariance. Furthermore, our empirical evidence indicates that models exhibiting consistent calibration across environments are also well-calibrated. This demonstrates that invariance and cross-environment calibration are empirically equivalent. Additionally, we underscore the necessity for a systematic approach to evaluating OOD generalization. This approach should move beyond traditional metrics, such as accuracy and F1 scores, which fail to account for the model's degree of over-confidence, and instead focus on the nuanced interplay between accuracy, calibration, and model invariance. Our code is available at https://github.com/katoro8989/IRM_Variants_Calibration

## 1 Introduction

In the evolving landscape of machine learning, the importance of out-of-distribution (OOD) generalization (Vapnik, 1991) and calibration (Guo et al., 2017) cannot be understated, especially in real-world applications and critical scenarios. Traditional machine learning approaches, grounded in the assumption that data are independently and identically distributed (IID), often struggle to cope with OOD scenarios. Additionally, a notable trend in contemporary machine learning models is their tendency towards overconfidence, which undermines the reliability of their confidence estimations. This issue is primarily recognized as a calibration challenge. Responding to these limitations, there has been a surge in research focusing on methodologies like Invariant Risk Minimization (IRM) (Arjovsky et al., 2020). IRM is designed to identify constant features across different environments, facilitating more robust generalization in the face of environmental

---

*equal contributions

variations. However, the computational demands of IRM have led to the exploration of more feasible, approximate methods. Despite the development of several IRM variants, their inability to consistently surpass the performance of finely-tuned Empirical Risk Minimization (ERM) models has been observed (Gulrajani & Lopez-Paz, 2020; Zhang et al., 2023a). This performance gap raises critical questions about the inherent deficiencies of these IRM variants. As we showed in Figure 1, even by leveraging IRM variants, there is a trade-off between in-distribution (ID) and OOD accuracy. This implies that models trained by IRM variants failed to obtain invariant features since they sacrificed ID accuracy to achieve better OOD accuracy. The invariance aimed for by IRM can, from another perspective, be conceptualized as the invariance of calibration degree, meaning that the model remains equally over-confident across all environments (see Section 2.3 for details). Our study intersects with this insight by evaluating the invariance of calibration through the variance of calibration metrics across environments, examining the extent to which IRM variants mimic the original model in terms of invariance. We hypothesize that an effective implementation of IRM should result in consistent calibration across all environments, indicated by minimal variance in calibration. The results reveal significant disparities in how calibration performance behaves across different environments depending on the IRM methodology used, even when OOD accuracy levels are similar. This suggests that relying solely on OOD accuracy for method development may be inadequate.

**Our key contributions** are as follows:

- We study discrepancies between IRM formulation and its implemented variants using the calibration metrics as a key metric to measure how much the model learned environmental invariant features.

- We observed that Information Bottleneck-based IRM (IB-IRM) improves the calibration performance across the environments (Figure 4 and Figure 2).

- We demonstrate that the trend of IB-IRM aligns more closely with the original objectives of IRM since its ECE variants across all environments are the smallest (Table 2).

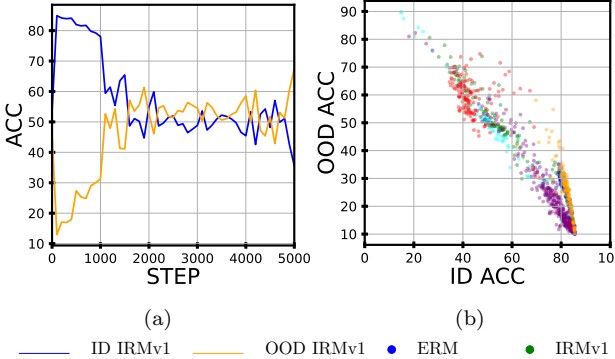

(a)       (b)

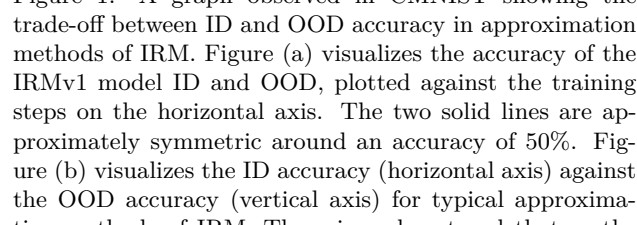

Figure 1: A graph observed in CMNIST showing the trade-off between ID and OOD accuracy in approximation methods of IRM. Figure (a) visualizes the accuracy of the IRMv1 model ID and OOD, plotted against the training steps on the horizontal axis. The two solid lines are approximately symmetric around an accuracy of 50%. Figure (b) visualizes the ID accuracy (horizontal axis) against the OOD accuracy (vertical axis) for typical approximation methods of IRM. There is a clear trend that as the accuracy improves OOD, the ID accuracy decreases.

## 2 Background

### 2.1 OOD Generalization

In traditional deep neural network scenarios, it is assumed that the learning and test environments are IID. The most commonly used learning algorithm in this assumption has been ERM (Vapnik, 1991), which has been successful in various scenarios. Specifically, ERM aims to minimize the sum of the risk of the model $f : \mathcal{X} \to \mathcal{Y}$ in the set of training environments $E_{train}$.

$$\min_{f:\mathcal{X} \to \mathcal{Y}} \sum_{e \in E_{train}} R^e(f) \tag{1}$$

where $R^e(f)$ is represented as $\mathbb{E}_{X^e, Y^e}[\ell(f(X^e), Y^e)]$, which is the risk in a environment $e$.

However, in the real world, the assumption of IID does not always hold, and distribution shifts between training and testing environments can occur (Quinonero-Candela et al., 2008). When the model relies on features that are only effective in the training environment, known as shortcut features, its performance can decrease in the testing environment (Geirhos et al., 2020). This issue is commonly known as OOD generalization and has become an urgent problem recently. Therefore, it is necessary to train robust models

that are not dependent on specific environments and can perform consistently across all environments. A model that is robust across the environment is defined as follows.

$$\mathbb{E}[Y^e|f(X^e)] = \mathbb{E}[Y^{e'}|f(X^{e'})], \forall e, e' \in E_{all} \tag{2}$$

where $E_{all}$ represents the set of all possible environments. Equation (2) indicates that, for a robust model, the conditional probability of the label given the model's prediction should be equal between any two environments.

Since it is difficult to access all real-world environments during training, various methods have been proposed to realize invariant models satisfying (eq. (2)) using limited learning environments.

### 2.2 Calibration

The alignment of a model's predictive confidence with its actual accuracy is extremely important in practical scenarios such as medical image diagnosis and autonomous driving. For example, if there are 100 instances where a model predicts with 80% confidence, then 80 of those predictions should be correct. However, in recent deep neural networks, issues such as increased model capacity have led to problems with miscalibration, and many methods have been proposed to address this calibration issue (Guo et al., 2017). The correct calibration in all environments is defined for any predictive probability $\alpha$ of $f$ as follows:

$$\mathbb{E}[Y^e|f(X^e) = \alpha] = \alpha, \forall e \in E_{all} \tag{3}$$

Equation (3) implies that the model's conditional predictive probability (in other words, it is called confidence) is always consistent with its actual accuracy.

**Expected Calibration Error**
The quality of uncertainty calibration is often quantified as ECE (Naeini et al., 2015) and is commonly used. It is defined as follows:

$$ECE = \sum_{m=1}^{M} \frac{|B_m|}{N}|Acc(B_m) - Conf(B_m)| \tag{4}$$

$Acc(B_m)$ and $Conf(B_m)$ are defined as follows

$$Acc(B_m) = \frac{1}{|B_m|} \sum_{b \in B_m} \mathbf{1}\left(\hat{y}_b = y_b\right), \ Conf(B_m) = \frac{1}{|B_m|} \sum_{b \in B_m} \hat{p}_b$$

where the predictive probabilities ranging from 0 to 1 are divided into $M$ parts, $Acc(B_m)$ is the prediction accuracy within the bin $B_m$ and $Conf(B_m)$ is the average of the prediction probability $\hat{p}_b$ (confidence level) for each data in the bin $B_m$.

**Adaptive Calibration Error**
Nixon et al. (2019) claimed that ECE is not perfect for measuring model calibration performance in some cases. Specifically, ECE calculates the discrepancy between predicted probabilities and actual accuracy for fixed bins based on predicted probabilities. If the model's predictions exhibit sharpness, this can result in a data imbalance within each bin. This trend of sharpness has been particularly noted in recent large-scale neural networks. The imbalance in the number of data points within bins implies a bias in the quality of calibration estimation, potentially leading to inaccurate assessments. To address the aforementioned issue, the Adaptive Calibration Error (ACE) has been proposed (Nixon et al., 2019). Unlike ECE, ACE divides the bins such that each bin contains the same number of samples, rather than being based on predicted probabilities. The formulation is as follows:

$$ACE = \frac{1}{KM} \sum_{k=1}^{K} \sum_{m=1}^{M} |Acc(B_{m,k}) - Conf(B_{m,k})| \tag{5}$$

where $K$ denotes the number of classes. The predicted probabilities corresponding to class $k$ for each data point are sorted and divided equally into $M$ bins, with the $m$-th bin denoted as $B_{m,k}$. In other words, each bin contains $\lfloor N/M \rfloor$ samples.

However, it should be noted that, in our scope of experiments, there is no significant difference between ECE and ACE results.

**Negative Log-Likelihood**
Additionally, the Negative Log-Likelihood (NLL) is commonly used as an indirect metric for measuring calibration performance (Minderer et al., 2021). The NLL is expressed by the following equation, and the closer the model's confidence in the correct label is to 1, the more favorable it is.

$$NLL = - \sum_{n=1}^{N} \log \hat{p}(y_n | x_n, \theta) \tag{6}$$

Therefore, even if the accuracy is 100%, improving the model's confidence can still enhance the NLL. If the model is overfitting on the NLL in the training domain and becoming overconfident, it is more likely to make incorrect predictions with overconfidence in the test domain. This leads to a higher NLL, indicating poor calibration performance (Mukhoti et al., 2020).

### 2.3 OOD Generalization and Calibration

Some empirical study (Naganuma & Hataya, 2023) reveals the correlation between OOD generalization and calibration. Theoretically, eq. (3) can be a sufficient condition for eq. (2), demonstrating that calibration across all environments leads to OOD generalization. Wald et al. (2021) capitalize on this observation and propose CLOvE, a method that calibrates models across multiple environments, thereby enabling generalization to unseen data distributions.

**Why do we use calibration to evaluate IRM variants?**
Since eq. (3) is a sufficient condition for eq. (2) and not a necessary one, it is possible for a model to achieve OOD while making overconfident predictions. Therefore, calibration performance does not necessarily perfectly explain OOD performance.

However, comparing the degree of calibration across different environments allows for measuring a model's invariance. The invariance condition is defined by the equality of the conditional probability $E[Y_e|f(X_e)]$ across all environments, which can be restated as the consistency of a model's miscalibration across all environments. Thus, evaluating the variance of a model's calibration metric across environments provides a means to assess its invariance. For example, if a model is consistently over-confident by 0.1 across all environments, meaning that $\mathbb{E}[Y^e|f(X^e) = \alpha] = \min(1.0, \alpha + 0.1) \ \forall e \in E_{all}$ holds, then the model satisfies invariance.

**How do we approximate invariance condition by using ECE?**
Each term of invariance condition eq. (2), can be approximated by the conditional accuracy, $\mathbb{E}[Y_e|f(X_e)] \approx Acc(B^e_m)$. Then, Equation (2) could be approximated as $Acc(B^e_m) = Acc(B^{e'}_m) \forall e, e' \in E_{all}$. And this equation can be easily transformed into $Conf(B^e_m) - Acc(B^e_m) = Conf(B^{e'}_m) - Acc(B^{e'}_m) \forall e, e' \in E_{all}$. Given that recent models are empirically overconfident (Minderer et al., 2021), we assume $Conf(B^e_m) > Acc(B^e_m)$, so the equation $Gap(B^e_m) = Conf(B^e_m) - Acc(B^e_m) = |Acc(B^e_m) - Conf(B^e_m)|$ holds true. Therefore, Equation (2) can be approximated as $ECE_e = ECE_{e'} \forall e, e' \in E_{all}$, and the consistency of ECE across environments can be used to approximate the model's invariance.

Furthermore, our empirical results indicate that cases of equivalent calibration failure between ID and OOD are rare. When calibration is consistent across domains, it is typically in well-calibrated cases. Hence,

empirical evidence suggests an equivalence between the invariance condition and calibration across all environments.

In this paper, we utilize this fact that comparing calibration performance across different environments will provide a better assessment of OOD generalization capabilities and evaluate the approximation methods of IRM, introduced in Section 4.

## 3 Preliminary

### 3.1 Domain Invariant Representation

One approach to making models more robust in OOD scenarios is through representation learning. Representation learning is a method that teaches the model to learn effective data representations from raw data for better performance in predictions. Traditionally, features of data were designed by human, but representation learning reduces this necessity and can also compress the dimensions of input data, enabling faster computations.

In the context of OOD scenarios, there is an application for representation learning, specifically aimed at extracting environment-invariant features from raw data and reducing environment-specific spurious features to achieve more robust predictions. For example, in the task of classifying animals from images, if the model learns to associate parts of an animal with extraneous elements, it may lead to high performance within the training domain but fail to make robust predictions when the background distribution shifts in the test domain. However, if the model discards spurious features (e.g., background) from the raw data, it can consistently make robust predictions across all environments. One particularly well-known method that applies representation learning to OOD generalization scenarios is Domain-Adversarial Neural Networks (DANN) (Ganin et al., 2016). This approach uses adversarial training to adjust the learning process so that it is impossible to discern from the extracted features which environment the input comes from, thereby aiming to extract invariant features.

### 3.2 Invariant Learning

In the context of applying representation learning to OOD generalization, the method we focus on in this paper is IRM (Arjovsky et al., 2020). IRM is proposed with the aim of extracting environment-invariant features from input data to enable consistent predictions across any environment. IRM is defined with the purpose of realizing (eq. (2)) as follows.

$$\min_{\substack{\Phi:\mathcal{X}\to\widehat{\mathcal{H}} \\ \omega:\widehat{\mathcal{H}}\to\widehat{\mathcal{Y}}}} \sum_{e\in E_{train}} R^e(\omega \circ \Phi) \tag{7}$$

$$subject\ to \quad \omega \in \arg\min_{\bar{\omega}:\widehat{\mathcal{H}}\to\widehat{\mathcal{Y}}} R^e(\bar{\omega} \circ \Phi), \forall e \in E_{train}$$

Here, $\mathcal{X}$ represents the input space of each data point, $\widehat{\mathcal{H}}$ denotes the extracted invariant feature space, and $\mathcal{Y}$ is the output space of the model. IRM defines the function as $f = \omega \circ \Phi$, where $\Phi$ maps the input data to $\widehat{\mathcal{H}}$, and based on these extracted invariant features, the final prediction is made by $\omega$. IRM aims to achieve consistent predictions as a form of OOD generalization.

The formulation in eq. (7) represents a bi-level optimization problem, which is challenging to solve exactly. As a result, various implementable approximation methods have been proposed. Many variants add various constraints to the IRM objective and approximate the bi-level optimization problem as a single-level problem to make it more tractable (Arjovsky et al., 2020; Ahuja et al., 2022; Chen et al., 2022; Ahuja et al., 2020; Lin et al., 2022). Additionally, Zhang et al. (2023b) propose an improved variant of IRM Game, another IRM variant, that retains the bi-level optimization framework during training. However, despite numerous contributions to the development of approximation methods for IRM, a method that generally surpasses well-tuned ERM in terms of OOD generalization accuracy has not yet been proposed. Therefore, it is essential

to understand why these approximation methods have not succeeded and what approaches might be more suitable for addressing the approximation problem.

While the original IRM formulation poses implementation challenges, an alternative method has also been proposed to address this issue. IRM aims to learn invariant features by keeping $\mathbb{E}[Y^e|f(X^e)]$ constant across environments. However, Huh & Baidya (2023) introduced a complementary notion of invariance called MRI, which seeks to conserve the label-conditioned feature expectation $\mathbb{E}[f(X^e)|Y^e]$ across environments, in contrast to the original IRM formulation.

## 4 Related Works

| Methods | Featurizer | Predictor | Linearity of $\omega$ | Information Bottleneck | Game theory | Ensemble | Bayesian inference |
|---|---|---|---|---|---|---|---|
| IRM | $\Phi$ | $\omega$ | ✗ | ✗ | ✗ | ✗ | ✗ |
| IRMv1 | $\omega \cdot \Phi$ | 1.0 | ✓ | ✗ | ✗ | ✗ | ✗ |
| IB-IRM | $\omega \cdot \Phi$ | 1.0 | ✓ | ✓ | ✗ | ✗ | ✗ |
| PAIR | $\omega \cdot \Phi$ | 1.0 | ✓ | ✗ | ✓ | ✗ | ✗ |
| IRM Game | $\Phi$ | $\omega_{ens}$ [1] | ✗ | ✗ | ✓ | ✓ | ✗ |
| BIRM | $\omega \cdot \Phi$ | 1.0 | ✓ | ✗ | ✗ | ✗ | ✓ |

Table 1: Approximation methods of IRM

### 4.1 IRMv1

IRMv1 (Arjovsky et al., 2020) constrains $\omega$ in eq. (7) as a linear mapping and allows the transformation $\Phi' = (\omega \cdot \Phi)$, $\omega' = 1.0$. The following one-variable optimization problem can be relaxed and computed.

$$\min_{\Phi':\mathcal{X} \to \mathcal{Y}} \sum_{e \in E_{train}} R^e(\Phi') \;+\; \lambda \parallel \nabla_{\omega'|\omega'=1.0} R^e(\omega' \cdot \Phi') \parallel^2 \tag{8}$$

### 4.2 Information Bottleneck based IRM(IB-IRM)

While IRMv1 has been successful with certain OOD generalization datasets such as CMNIST, it has been noted that if not enough invariant features are included in each environment in the training data, robust prediction is lost. Therefore, IB-IRM (Ahuja et al., 2022) aims to improve IRMv1 by compressing the entropy of the feature extractor $\Phi'$ using the information bottleneck method, thereby preventing excessive reliance on features that adversely affect the invariant prediction. The variance of the features extracted from each data by $\Phi'$ is added as a regularization term.

$$\lambda \parallel \nabla_{\omega'|\omega'=1.0} R^e(\omega' \cdot \Phi') \parallel^2 + \gamma Var(\Phi') \tag{9}$$

### 4.3 Pareto IRM(PAIR)

In general, there is a trade-off between ERM and OOD generalization, and PAIR (Chen et al., 2022) focuses on the need to properly manage that trade-off when generalizing a model out of distribution. To achieve more robust models, PAIR is designed to find their Pareto optimal solutions in terms of multi-objective optimization. In fact, it is implemented as a multi-objective optimal problem for ERM, IRMv1 penalties, and VREx (Krueger et al., 2021) penalties.

---

[1] $\omega_{ens} = \frac{1}{|E_{train}|} \left( \sum_{e \in E_{train}} w^e \right)$

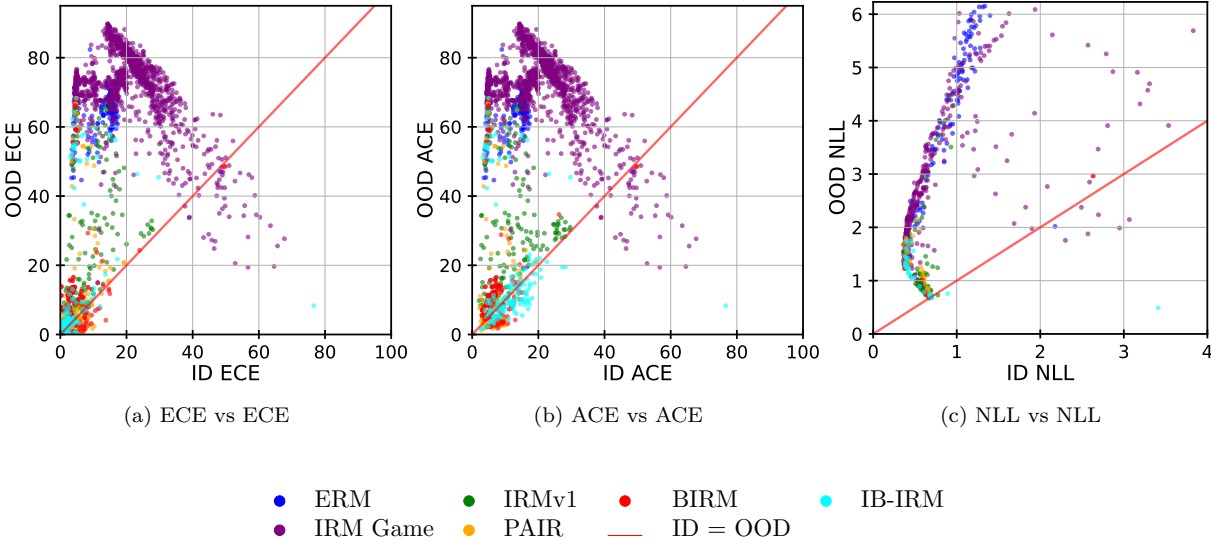



● ERM    ● IRMv1    ● BIRM    ● IB-IRM

● IRM Game    ● PAIR    —— ID = OOD



Figure 2: ECE, ACE, and NLL evaluation on the CMNIST. The X-axis indicates the evaluation of ID sets, and the Y-axis indicates the evaluation of OOD sets. The red solid line represents the case when the metric values are equal for both ID and OOD, indicating that the model has achieved the same level of calibration performance in both domains. IRMv1, IB-IRM, BIRM, and PAIR are distributed relatively close to the red solid line, indicating that they achieve consistent calibration across environments. Those data points that show better OOD calibration performance for each IRM variant tend to be closer to the red line.

$$\min_{\Phi':\mathcal{X}\to\mathcal{Y}}(\mathcal{L}_{ERM},\mathcal{L}_{IRMv1},\mathcal{L}_{VREx}) \tag{10}$$

where

$$\mathcal{L}_{ERM}=\sum_{e\in E_{train}}R^e(\Phi'),\ \ \mathcal{L}_{IRMv1}=\parallel\nabla_{\omega'|\omega'=1.0}R^e(\omega'\cdot\Phi')\parallel^2,\ \ \mathcal{L}_{VREx}=Var(\{R^e(\Phi')\}_{e\in E_{train}})$$

### 4.4 IRM Game

IRM Game (Ahuja et al., 2020) aims to achieve Nash equilibrium regarding inference accuracy by introducing a game-theoretic framework to the IRM between each environment. Each environment $e$ in the training data is assigned its own predictor $\omega^e$, each of which is trained to be optimal in each environment, and the final $\omega_{ens}$ is the ensemble of all predictors $\frac{1}{|E_{train}|}\sum_{e\in E_{train}}\omega^e$. By learning environment-specific predictors, the limitation on linearity made in IRMv1 is eliminated, and implementation closer to eq. (7) is aimed at.

$$\min_{\substack{\Phi:\mathcal{X}\to\widehat{\mathcal{H}}\\\omega_{ens}:\widehat{\mathcal{H}}\to\widehat{\mathcal{Y}}}}\sum_{e\in E_{train}}R^e(\omega_{ens}\circ\Phi) \tag{11}$$

$$s.t.\ \omega^e\in\arg\min_{\bar{\omega}^e:\widehat{\mathcal{H}}\to\widehat{\mathcal{Y}}}R^e\left\{\bar{\omega}_{ens}^e\circ\Phi\right\},\forall e\in E_{train}$$

where $\bar{\omega}_{ens}^e=\frac{1}{|E_{train}|}\left(\bar{\omega}^e+\sum_{e'\neq e}w^{e'}\right)$

### 4.5 Bayesian IRM(BIRM)

In BIRM (Lin et al., 2022), IRMv1 may overfit the training environment if the training data are insufficient, so we introduced a Bayesian estimation approach there. Noting that the posterior probability $p(\omega^e|\Phi(X^e),Y^e)$

is invariant in all environments given the data mapped by $\Phi$ and the correct label if the model is able to acquire invariant features, the penalty of IRMv1 is modified. By using Bayesian estimation instead of point estimation, they aim to improve the generalization performance of the model by taking into account the uncertainty of the data.

Table 1 presents an overview of IRM and its approximation methods.

## 5 Experiments

Although the approximation methods in the previous chapter have achieved some success in terms of OOD accuracy on OOD datasets (Arjovsky et al., 2020; Ahuja et al., 2022; Chen et al., 2022; Ahuja et al., 2020; Lin et al., 2022), it is not known how invariant features they actually acquire and how robust they are to unknown environments. Therefore, we evaluated them by comparing their calibration performance with ECE, ACE, and NLL. Since we aim to investigate OOD performance through invariance condition eq. (2), we do not use some metrics that do not incorporate the model's conditional probability, such as the F1 score.

### 5.1 Setting

We compared ECE of each model on four different OOD-datasets. Specifically, the datasets used were Colored MNIST (CMNIST) (Arjovsky et al., 2020), Rotated MNIST (RMNIST) (Ghifary et al., 2015), PACS (Li et al., 2017), and VLCS (Fang et al., 2013), sourced from the DomainBed benchmark (Gulrajani & Lopez-Paz, 2020). These datasets can be distinguished based on the type of distribution shift they represent (Ye et al., 2022). The distribution shift in CMNIST is known as the correlation shift, where the conditional probability of labels given spurious features changes across environments. The other three datasets—RMNIST, PACS, and VLCS—are classified under diversity shift, where the prior probability of spurious features changes across environments.

For optimization, Adam (Kingma & Ba, 2015) was used consistently across all models, and the tuning of learning rates and hyperparameters for each method was conducted in accordance with their respective papers.

### 5.2 Correlation Shift

#### 5.2.1 Dataset

CMNIST is a dataset based on MNIST, containing 70,000 dimension examples (1, 28, 28). It is adapted for a binary classification task where digits less than 5 are labeled as class 0, and those 5 or above as class 1. Spurious correlations are added by assigning colors (red or green) to the digits, with the proportion of the two colors varying in each class depending on the environment. Denoting the proportion of green in class 0 as environment $e$, then the training environments are set as $e = [10\%, 20\%]$, and the test environment as $e = [90\%]$.

#### 5.2.2 Results

Figure 2 presents a scatter plot showing the relationship between evaluation on ID and that on OOD for multiple IRM variants, each trained with multiple hyperparameters. The red solid line represents the case when the metric values are equal for both ID and OOD, indicating that the model has achieved the same level of calibration performance in both domains. IRMv1 (green), IB-IRM (light blue), BIRM (red), and PAIR (yellow) are distributed near the red solid line. Additionally, they exhibit better calibration performance in both domains. On the other hand, ERM (blue) and IRM Game (purple) are distributed above the red solid line, showing worse calibration performance in OOD. This suggests that they may be overfitting to the training domain, resulting in overconfidence in the test domain.

In some cases, IRM Game shows poor calibration performance but still appears near the red solid line. However, it does not exhibit stable distribution. This instability is likely due to the presence of separate

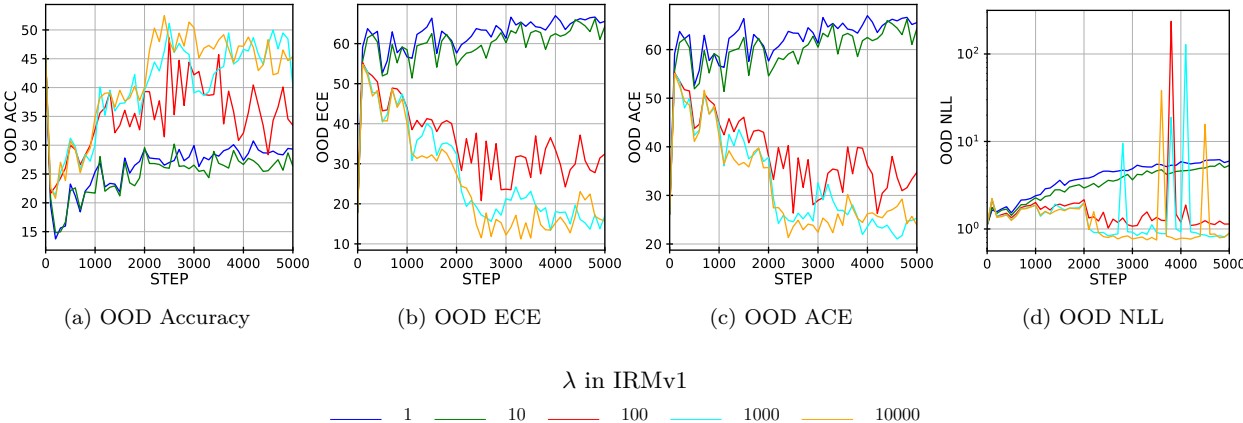

Figure 3: Comparison of the model's OOD calibration performance in CMNIST, with various values of $\lambda$ in the formulation of IRMv1 eq. (8). It was observed that increasing $\lambda$, thereby increasing the regularization penalty in IRMv1, results in improved calibration performance for the model in the OOD environment.

classifiers $\omega^e$ fitted for each environment, which may lead to overfitting within each environment, making it difficult to balance across them.

Empirically, cases, where calibration performance is consistent between ID and OOD, are limited to those where both domains exhibit better calibration performance. Specifically, as shown in Figure 2c, instances distributed along the red solid line are confined to points with low NLL in both domains. This trend is also observed in ECE (Figure 2a) and ACE (Figure 2b), suggesting that, empirically, cross-environment calibration is a necessary condition for invariance.

Moreover, Figure 3 visualizes how the penalty introduced in IRMv1 for the purpose of OOD generalization actually affects the calibration performance. It specifically compares different values of $\lambda$ in IRMv1's formulation eq. (8), demonstrating that as the impact of IRMv1's penalty increases, the calibration performance improves. It was also found that the information bottleneck penalty in IB-IRM is effective in improving calibration. Detailed results are presented in Figure 7 and Figure 8 in Appendix B.2.

### 5.3 Diversity Shift

#### 5.3.1 Datasets

The details for RMNIST, PACS, and VLCS are as follows:

**RMNIST:** RMNIST is a dataset based on MNIST, containing 70,000 examples of the same dimension (1, 28, 28) and includes 10 classes. The environments are distinguished by the rotation angle of the digits, denoted as $e$. The training environments are set at $e = [15°, 30°, 45°, 60°, 75°]$, while the test environment is at $e = [0°]$.

**PACS:** PACS is a dataset containing 9,991 examples with dimensions (3, 224, 224) and includes 7 classes. The environments are distinguished by four different styles of images. Specifically, the training environments are set as $e = [Cartoons, Photos, Sketches]$, and the test environment is set as $e = [Art]$.

**VLCS:** VLCS is a dataset containing 10,729 examples with dimensions (3, 224, 224) and includes 5 classes. There are four types of environments. The training environments are set as $e = [Caltech101, LabelMe, SUN09]$, and the test environment is set as $e = [VOC2007]$.

#### 5.3.2 Results

Considering that a model's OOD accuracy affects its calibration (Ovadia et al., 2019), we compared the calibration performance with OOD accuracy for a fairer comparison of the calibration metrics. A threshold was set for validation accuracy in the training environment, and upon reaching this threshold, early stopping

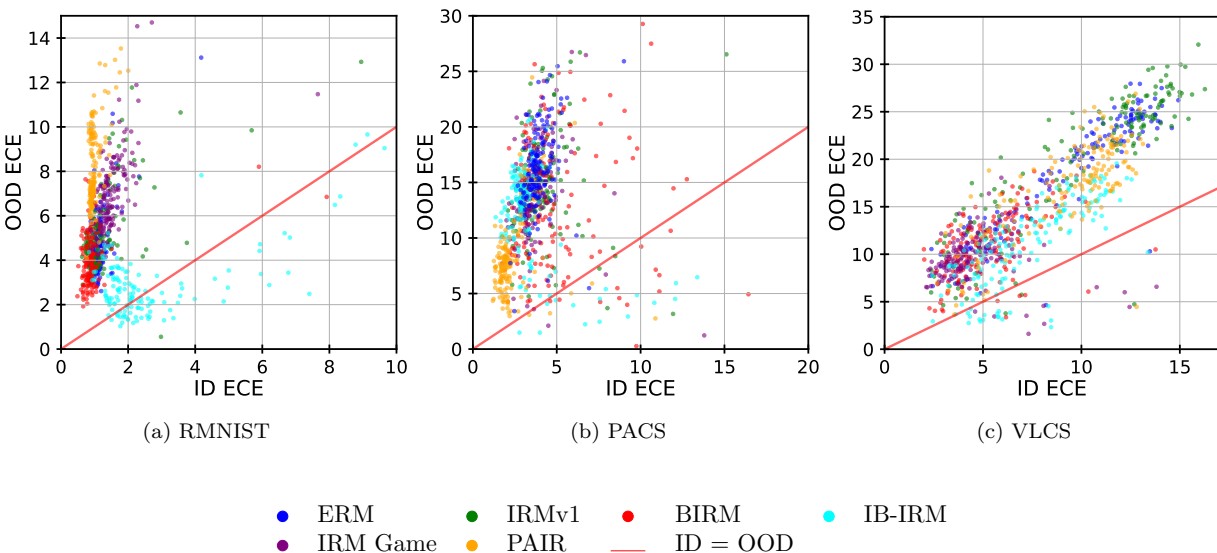

Figure 4: Comparison of the relationship between ECE in the training environment (horizontal axis) and the test environment (vertical axis). The red solid line represents the case where the ECE is equal in both environments. It was observed that IB-IRM (in light blue) is distributed near the red solid line, indicating a tendency not to overfit to the training environment compared to other methods.

of training was implemented to create conditions of comparable OOD accuracy. The threshold for early stopping was selected using grid search to minimize the variance in test accuracy across different methods.

Table 2 shows the variance in calibration performance across all environments for each method on different datasets. Despite similar OOD accuracy, the calibration performance behavior varies across methods. Notably, IB-IRM consistently achieves a lower variance in most cases, indicating that it maintains similar calibration across all environments and achieves invariance. In Figure 4, it is also evident that IB-IRM achieves similar calibration performance for both ID and OOD. Additional ACE and NLL results are presented in Appendix B.1.

Similar to the Correlation Shift, it was observed that when calibration performance is consistent across both domains, there is a strong tendency to achieve good calibration performance. Detailed explanations along with more experimental results are provided in Appendix B.1.

## 6 Information Bottleneck in Calibration

In our experiments, we observe that IRM with information bottleneck is effective in calibration, as shown in Figure 2 and Figure 4. Therefore, in this section, we provide the results of the ablation study for IB-IRM.

We investigated the impact of the penalty term related to the information bottleneck in IB-IRM on calibration. Figure 5 visualizes various evaluation metrics in CMNIST when altering the strength of the information bottleneck in IB-IRM, that is, the magnitude of $\gamma$ in eq. (9). As shown in the top row of Figure 5, which visualizes this based on OOD calibration and OOD accuracy, we confirm the clear trend that strengthening the regularization of $\gamma$ improves both accuracy and ECE. The bottom row of Figure 5 shows the relationship between the calibration metrics in ID and OOD. Increasing $\gamma$ aligns both calibration metrics towards the red line, representing that this constitutes a successful calibration across environments. These findings suggest that not only is the IRMv1's regularization term aimed at OOD scenarios, but the information bottleneck also further promotes the learning of invariant features.

As an intuitive explanation of the information bottleneck approach's benefit for ECE is as follows:

1. The information bottleneck explicitly regularizes entropy.
2. This leads to the prevention of overfitting to one-hot encoded labels.
3. As a result, the model trained by information bottleneck is well calibrated

|  |  | ERM | IRMv1 | IB-IRM | BIRM | IRM Game | PAIR |
|---|---|---|---|---|---|---|---|
| OOD ACC | RMNIST | $88.3_{\pm0.1}$ | $89.7_{\pm2.3}$ | $85.4_{\pm1.3}$ | $87.6_{\pm1.1}$ | $87.8_{\pm1.4}$ | $82.7_{\pm1.4}$ |
|  | PACS | $73.8_{\pm3.9}$ | $76.9_{\pm1.9}$ | $75.5_{\pm0.8}$ | $67.2_{\pm1.7}$ | $72.7_{\pm1.8}$ | $76.0_{\pm1.6}$ |
|  | VLCS | $70.0_{\pm0.8}$ | $67.7_{\pm2.0}$ | $69.5_{\pm0.5}$ | $70.2_{\pm2.0}$ | $69.5_{\pm0.7}$ | $67.9_{\pm0.2}$ |
|  | Avg. | $77.4_{\pm1.6}$ | $78.1_{\pm2.1}$ | $76.8_{\pm0.9}$ | $75.0_{\pm1.6}$ | $76.7_{\pm1.3}$ | $75.5_{\pm1.1}$ |
| ECE Variance | RMNIST | $2.21_{\pm0.97}$ | $3.61_{\pm1.73}$ | $\mathbf{0.17_{\pm0.03}}$ | $1.85_{\pm0.49}$ | $4.37_{\pm1.79}$ | $11.17_{\pm3.04}$ |
|  | PACS | $19.86_{\pm17.38}$ | $23.62_{\pm1.77}$ | $22.10_{\pm6.39}$ | $45.78_{\pm14.01}$ | $16.73_{\pm6.05}$ | $\mathbf{6.19_{\pm7.49}}$ |
|  | VLCS | $17.97_{\pm2.51}$ | $65.99_{\pm5.82}$ | $\mathbf{5.80_{\pm3.41}}$ | $12.84_{\pm5.57}$ | $6.49_{\pm2.46}$ | $33.50_{\pm19.74}$ |
|  | Avg. | $13.35_{\pm6.95}$ | $31.07_{\pm3.11}$ | $9.36_{\pm3.28}$ | $20.16_{\pm6.69}$ | $\mathbf{9.22_{\pm27.59}}$ | $16.95_{\pm10.09}$ |
| ACE Variance | RMNIST | $2.31_{\pm0.87}$ | $3.68_{\pm1.79}$ | $\mathbf{0.18_{\pm0.08}}$ | $1.85_{\pm0.48}$ | $4.37_{\pm2.04}$ | $11.65_{\pm2.96}$ |
|  | PACS | $21.15_{\pm18.69}$ | $25.33_{\pm1.42}$ | $20.24_{\pm7.27}$ | $47.31_{\pm13.65}$ | $18.94_{\pm5.86}$ | $\mathbf{6.29_{\pm6.73}}$ |
|  | VLCS | $20.20_{\pm2.23}$ | $67.65_{\pm6.06}$ | $\mathbf{6.29_{\pm3.59}}$ | $12.29_{\pm4.67}$ | $6.98_{\pm1.88}$ | $33.90_{\pm19.90}$ |
|  | Avg. | $14.55_{\pm7.26}$ | $32.22_{\pm3.09}$ | $\mathbf{8.90_{\pm3.65}}$ | $20.48_{\pm6.27}$ | $10.10_{\pm3.26}$ | $17.28_{\pm9.86}$ |
| NLL Variance | RMNIST | $\mathbf{0.01_{\pm0.00}}$ | $0.02_{\pm0.01}$ | $0.02_{\pm0.00}$ | $0.02_{\pm0.00}$ | $0.03_{\pm0.02}$ | $0.05_{\pm0.01}$ |
|  | PACS | $0.15_{\pm0.13}$ | $0.12_{\pm0.04}$ | $0.09_{\pm0.01}$ | $0.23_{\pm0.06}$ | $0.12_{\pm0.01}$ | $\mathbf{0.05_{\pm0.01}}$ |
|  | VLCS | $0.12_{\pm0.01}$ | $0.32_{\pm0.01}$ | $\mathbf{0.10_{\pm0.01}}$ | $0.11_{\pm0.01}$ | $\mathbf{0.10_{\pm0.01}}$ | $0.19_{\pm0.03}$ |
|  | Avg. | $0.09_{\pm0.05}$ | $0.15_{\pm0.02}$ | $\mathbf{0.07_{\pm0.01}}$ | $0.12_{\pm0.02}$ | $0.08_{\pm0.01}$ | $0.10_{\pm0.02}$ |

Table 2: Comparison of calibration performance consistency across environments for RMNIST, PACS, and VLCS. This analysis presents the variance in ECE, ACE, and NLL across environments, where smaller variances indicate more consistent calibration across environments. Each model was trained with hyperparameter tuning using three different seeds, and the average variance across environments was calculated. The results show that IB-IRM achieved the smallest average variance, indicating that it consistently maintains calibration across different environments.

It is known that the reason for the degradation of calibration performance in recent large-scale neural networks is due to the overconfidence caused by their complexity (Guo et al., 2017). By keeping entropy low and maintaining the model relatively simple, overconfidence is prevented, which explains the superior calibration performance of the information bottleneck.

As a technical explanation, we describe the validity of our results based on the theoretical principles of the information bottleneck. In OOD scenarios, the objective of the information bottleneck is to minimize the mutual information $I(X; \Phi(X))$ between the input $X$ and the intermediate representation $\Phi(X)$ while preserving the mutual information $I(\Phi(X); Y)$ between $\Phi(X)$ and the output $Y$. Intuitively, this means learning $\Phi(X)$ to reduce as much information from $X$ as possible while retaining as much information about $Y$. When $\Phi(X)$ is a deterministic transformation of $X$, the entropy $H(\Phi(X))$ can be used to minimize $I(X; \Phi(X))$ (Kirsch et al., 2021). In the context of IRM, which aims to learn invariant features, keeping $H(\Phi(X))$ small encourages $\Phi$ to capture invariant features $X_{inv}$ from $X$ and discard spurious features $X_{sup}$. It should be noted that Ahuja et al. (2022) theoretically demonstrated, using realistic SEMs, that minimizing $H(\Phi(X))$ in IRM leads to $\Phi$ discarding spurious features and grasping invariant features.

Our results empirically support the effectiveness of applying the information bottleneck method to invariant feature learning.

# 7 Discussion and Conclusion

In this paper, we focus on IRM, which is challenging to implement due to the difficulties of bi-level optimization. We conducted comparative evaluations using ECE, ACE, and NLL, computationally feasible metrics for calibration, to understand its approximation methods better and measure how much the model learned environmental invariant features.

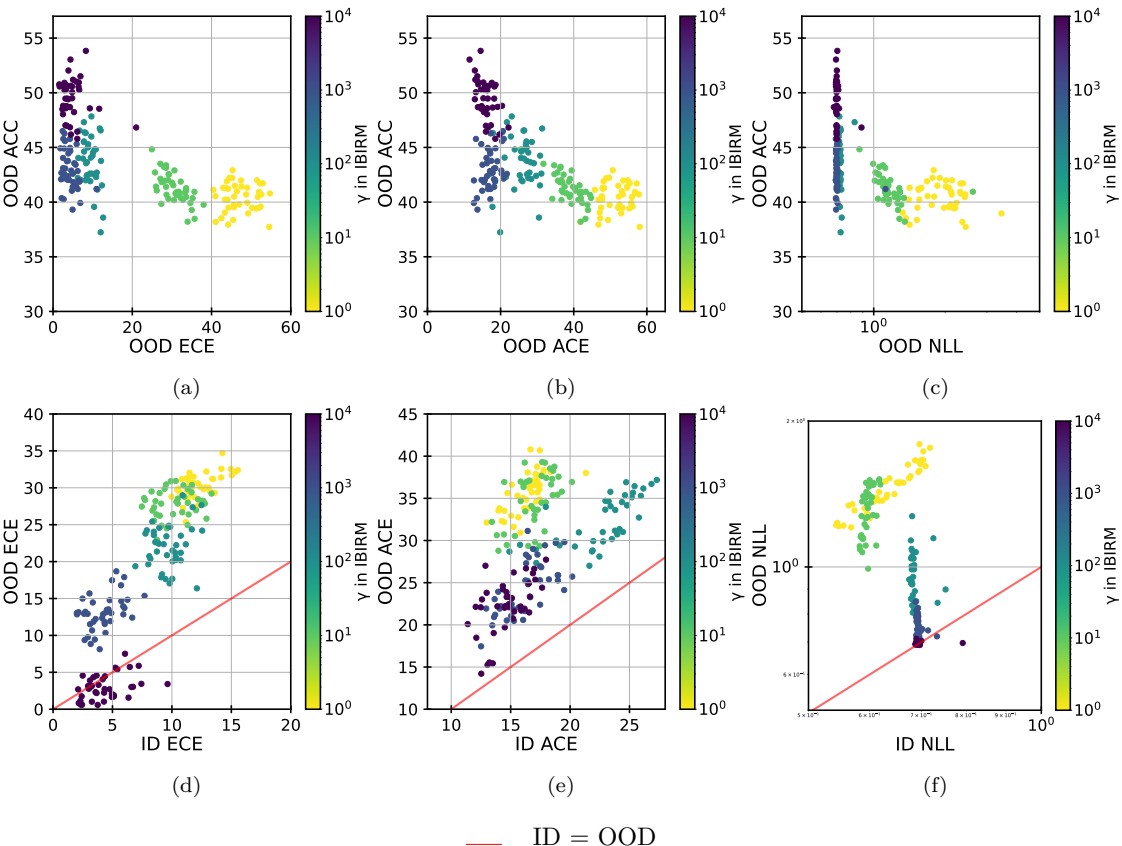

Figure 5: Impact of the information bottleneck on calibration with IB-IRM on CMNIST task. The top row displays the relationship between calibration metrics and Accuracy in the OOD context, investigated by varying the coefficient $\gamma$ of the information bottleneck penalty in the IB-IRM formulation (eq. (9)). As the value of $\gamma$ increases, both metrics show improvement. The bottom row visualizes the values of calibration metrics in both ID and OOD. Altering the value of $\gamma$ shows that the larger the value, the more the data aligns with the red line, which indicates equality between the two calibration performances. With a sufficiently large $\gamma$, the points are almost perfectly distributed along the red line, indicating successful calibration across multiple environments.

First, our empirical results suggest that calibration performance is equivalent across environments only when calibration performance is relatively high. This suggests that calibration across environments is not only a sufficient condition for invariance, but also a necessary condition empirically, making it clear that calibration is extremely effective for learning invariant features.

It is also suggested that IB-IRM, which applies the information bottleneck method to IRM, achieves more consistent and higher calibration under dataset shift, indicating that it performs invariant feature learning better than other variants.

Another finding from our empirical results is that regularization penalties introduced for OOD generalization effectively improve calibration. However, in more practical datasets (under diversity shift), we observe a trade-off between OOD calibration performance and accuracy. Although IB-IRM mitigates this tradeoff compared to IRMv1 and other variants; carefully balancing these two metrics is important in aiming for OOD generalization.

As a future challenge, it has become clear that the degree of invariant feature learning cannot be fully measured by accuracy on OOD data alone, necessitating the establishment of systematic generalization metrics and measurement methods. Furthermore, a better understanding of the relationship between OOD generalization and uncertainty calibration, as well as the development of novel approximation methods for IRM based on this understanding, is anticipated. In particular, gaining a deeper understanding of the impact

of the information bottleneck on OOD generalization and calibration will be key to addressing the issues of OOD generalization and calibration. A more comprehensive grasp of how the information bottleneck principle affects the model's ability to generalize to unseen environments and maintain well-calibrated predictive uncertainties will be crucial in developing more robust and reliable models.

## Acknowledgments

We acknowledge the generous allocation of computational resources from the TSUBAME3.0 supercomputer facilitated by the Tokyo Institute of Technology. This assistance came through the TSUBAME Encouragement Program for Young / Female Users, whose support was instrumental to this research.

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

# Appendix

## A    Details of experiments

### A.1    Implementation of IRM Variants

This section provides a detailed description of each method used in the experiments.

#### A.1.1    IRMv1

The learning process was conducted using the loss calculated based on the formulation shown in 8. The hyperparameter $\lambda$ is used to adjust the impact of the IRMv1 penalty.

#### A.1.2    IBIRM

The learning process was performed using the loss calculated based on the formulation shown in 9. The hyperparameters $\lambda$ and $\gamma$ are used to adjust the influence of the IRMv1 penalty and the information bottleneck penalty, respectively.

#### A.1.3    IRM Game

IRM Game has two variants: F-IRM Game and V-IRM Game. The difference between them lies in whether the featurizer $\Phi$ is fixed or not. F-IRM Game fixes $\Phi = I$ as an identity matrix, while V-IRM Game considers a variable $\Phi$. For each dataset, the variant that achieved higher performance in terms of accuracy was selected.

#### A.1.4    PAIR

PAIR employs a weighted average of ERM loss, IRMv1 loss, and VREx loss as the final loss for learning. When calculating the weights, a hyperparameter called *preference* is used to adjust the scaling of each loss.

#### A.1.5    BIRM

In BIRM, when calculating the IRMv1 loss, Monte Carlo sampling is used to sample $N$ times from the posterior distribution of the parameters, and the average of these samples is used to compute the loss. This approach takes into account the uncertainty of the parameters when calculating the loss. The standard deviation of the Gaussian noise added to the parameters during sampling is treated as a hyperparameter denoted as *birm_sd*. In all experiments, $N$ was set to 5.

### A.2    Datasets

#### A.2.1    Domainbed

We conducted experiments using Colored MNIST (CMNIST) (Arjovsky et al., 2020), Rotated MNIST (RMNIST) (Ghifary et al., 2015), PACS (Li et al., 2017), and VLCS (Fang et al., 2013) datasets from DomainBed (Gulrajani & Lopez-Paz, 2020). We split each dataset into training and validation sets, with 80% used for training and the remaining 20% for validation. The environment partitions were as follows:

- CMNIST: $E_{train} = [10\%, 20\%]$, $E_{test} = [90\%]$

- RMNIST: $E_{train} = [15°, 30°, 45°, 60°, 75°]$, $E_{test} = [0°]$

- PACS: $E_{train} = [Photo, Painting, Sketch]$, $E_{test} = [Art]$

- VLCS: $E_{train} = [Caltech101, LabelMe, SUN09]$, $E_{test} = [VOC2007]$

### A.2.2 Hyperparameters

In the experiments, we set the batch size to 256 for CMNIST, 128 for RMNIST, and 16 for PACS and VLCS. Grid search was performed on the learning rate for all experiments, with values of [1e-4, 5e-4, 1e-3, 5e-3]. For the hyperparameters specific to each approximation method, grid search was conducted as shown in Table 3.

| Parameter | |
|---|---|
| $\lambda$ | [1, 1e1, 1e2, 1e3, 1e4] |
| $\gamma$ | [1, 1e1, 1e2, 1e3, 1e4] |
| $preference$ | [0, 1, 2, 3, 4] |
| $birm\_sd$ | [5e-2, 1e-1, 2e-1] |

Table 3: Hyperparameters for each method

Model selection was performed using an oracle based on accuracy in the test environment for Correlation shift, while for Diversity shift, it was based on validation accuracy in the training environment. Correlation shift is known to be a relatively challenging task(Ye et al., 2022) as it involves a shift in the conditional probability $P[Y|X]$, where $X$ represents the input data and $Y$ represents the corresponding ground truth labels. As shown in Figure 1(b), there exists a trade-off between accuracy in the training and test environments for all methods, necessitating oracle model selection. In contrast, for Diversity shift, there is a correlation between ID and OOD accuracy across all methods as shown in Figure 10, allowing for model selection based on performance in the training environment.

## B Additional results

### B.1 Ablation Study on Diversity Shift

First, Table 4, similar to Table 2, compares the calibration performance of each method under the same early stopping settings. While Table 2 compares the variance in calibration performance across environments, Table 4 provides a straightforward comparison of the calibration metrics. Although the methods achieve similar OOD accuracy, their calibration performance varies, with IB-IRM achieving consistently better average calibration performance.

Next, Figure 4 visualizes the relationship between ID and OOD calibration performance for each dataset. The red solid line represents cases where calibration performance is equal in both domains, indicating successful cross-environment calibration. Notably, IB-IRM (light blue) is distributed relatively close to the red solid line in RMNIST and VLCS, while PAIR (yellow) is close to the red line in PACS. This suggests that these methods achieve consistent calibration across environments.

Combined with the results in Table 2, it can be concluded that IB-IRM consistently achieves high calibration performance across environments on average.

Additionally, from the plots in Figure 4, a trend can be observed where lower calibration performance tends to be closer to the red solid line. This implies that achieving consistent calibration across environments, in other words, achieving invariance, requires higher calibration performance.

### B.2 Impact of penalties of IBIRM

Figure 3 in CMNIST compares the OOD ECE of the model when varying the regularization penalty $\lambda$ in IRMv1. Additionally, we compare the OOD ECE of the model when varying the coefficients of the two regularization penalties in IBIRM Equation (9) as shown in Figure 7 and Figure 8. Similar to Figure 3, as the regularization penalty increases, the OOD ECE decreases.

| | | ERM | IRMv1 | IB-IRM | BIRM | IRM Game | PAIR |
|---|---|---|---|---|---|---|---|
| **OOD ACC** | RMNIST | $88.3_{\pm0.1}$ | $89.7_{\pm2.3}$ | $85.4_{\pm1.3}$ | $87.6_{\pm1.1}$ | $87.8_{\pm1.4}$ | $82.7_{\pm1.4}$ |
| | PACS | $73.8_{\pm3.9}$ | $76.9_{\pm1.9}$ | $75.5_{\pm0.8}$ | $67.2_{\pm1.7}$ | $72.7_{\pm1.8}$ | $76.0_{\pm1.6}$ |
| | VLCS | $70.0_{\pm0.8}$ | $67.7_{\pm2.0}$ | $69.5_{\pm0.5}$ | $70.2_{\pm2.0}$ | $69.5_{\pm0.7}$ | $67.9_{\pm0.2}$ |
| | Avg. | $77.4_{\pm1.6}$ | $78.1_{\pm2.1}$ | $76.8_{\pm0.9}$ | $75.0_{\pm1.6}$ | $76.7_{\pm1.3}$ | $75.5_{\pm1.1}$ |
| **OOD ECE** | RMNIST | $5.01_{\pm1.07}$ | $5.74_{\pm1.42}$ | $\mathbf{2.40_{\pm0.64}}$ | $4.45_{\pm0.42}$ | $6.71_{\pm1.09}$ | $9.55_{\pm0.80}$ |
| | PACS | $13.05_{\pm5.21}$ | $12.97_{\pm2.15}$ | $11.98_{\pm0.43}$ | $18.61_{\pm1.80}$ | $12.16_{\pm1.52}$ | $\mathbf{7.03_{\pm3.32}}$ |
| | VLCS | $12.55_{\pm1.39}$ | $22.79_{\pm0.85}$ | $\mathbf{7.97_{\pm2.29}}$ | $9.94_{\pm2.41}$ | $8.78_{\pm1.23}$ | $17.79_{\pm2.96}$ |
| | Avg. | $10.20_{\pm2.56}$ | $13.83_{\pm1.47}$ | $\mathbf{7.45_{\pm1.12}}$ | $11.00_{\pm1.54}$ | $9.22_{\pm1.28}$ | $11.45_{\pm2.36}$ |
| **OOD ACE** | RMNIST | $4.94_{\pm1.06}$ | $5.65_{\pm1.43}$ | $\mathbf{2.26_{\pm0.59}}$ | $4.40_{\pm0.42}$ | $6.56_{\pm1.20}$ | $9.49_{\pm0.82}$ |
| | PACS | $12.99_{\pm5.22}$ | $12.91_{\pm2.09}$ | $11.86_{\pm0.41}$ | $18.53_{\pm1.84}$ | $12.06_{\pm1.61}$ | $\mathbf{6.74_{\pm3.33}}$ |
| | VLCS | $12.48_{\pm1.37}$ | $22.71_{\pm0.82}$ | $\mathbf{8.07_{\pm2.16}}$ | $9.88_{\pm2.37}$ | $8.72_{\pm1.18}$ | $17.79_{\pm2.96}$ |
| | Avg. | $10.14_{\pm2.55}$ | $13.76_{\pm1.45}$ | $\mathbf{7.40_{\pm1.05}}$ | $10.94_{\pm1.54}$ | $9.11_{\pm1.33}$ | $11.34_{\pm2.37}$ |
| **OOD NLL** | RMNIST | $\mathbf{0.41_{\pm0.03}}$ | $0.43_{\pm0.10}$ | $0.48_{\pm0.03}$ | $0.42_{\pm0.04}$ | $0.54_{\pm0.11}$ | $0.66_{\pm0.04}$ |
| | PACS | $1.07_{\pm0.36}$ | $1.00_{\pm0.19}$ | $0.87_{\pm0.03}$ | $1.34_{\pm0.11}$ | $1.06_{\pm0.07}$ | $\mathbf{0.75_{\pm0.05}}$ |
| | VLCS | $0.97_{\pm0.08}$ | $1.63_{\pm0.03}$ | $\mathbf{0.88_{\pm0.04}}$ | $0.90_{\pm0.08}$ | $0.89_{\pm0.04}$ | $1.25_{\pm0.11}$ |
| | Avg. | $0.82_{\pm0.16}$ | $1.02_{\pm0.11}$ | $\mathbf{0.74_{\pm0.03}}$ | $0.89_{\pm0.08}$ | $0.83_{\pm0.07}$ | $0.89_{\pm0.07}$ |

Table 4: Comparison of OOD accuracy and the calibration metrics across RMNIST, PACS, and VLCS datasets. Following hyperparameter tuning for each method, measurements were taken across three different random seeds, presenting the mean and variance for each metric. It was observed that on average, IB-IRM exhibits a lower the calibration metrics while maintaining comparable inference performance.

### B.3 Feature visualization study

To investigate how differences in ECE impact the model's ability to learn invariant features, we conducted a feature visualization study. Using a toy CNN, we visualized the latent features of IRMv1 and IB-IRM on RMNIST. The visualizations were created by applying t-SNE (Van der Maaten & Hinton, 2008) to the outputs of the final convolutional layer of the CNN. The results are shown in Figure 9.

First, the table presents the variance in ECE across the six environments of RMNIST, indicating that IB-IRM achieves more stable calibration. Next, consider the four figures. The top row visualizes the feature distributions of each model in the ID setting, while the bottom row shows the OOD setting. The color of each point represents the class, as RMNIST involves 10-class classification.

In a truly successful invariant feature learning scenario, the distributions should be similar in both the top and bottom rows. IB-IRM (right column) shows stable class feature distributions in both ID and OOD. In contrast, IRMv1, with its higher ECE variance, displays variations in these distributions, indicating that the features it extracts differ depending on the environment. This study demonstrates that stable calibration performance across environments reflects true invariance.

### B.4 Relationship between ID and OOD accuracy in diversity shift

As shown in Figure 1(b), in the presence of correlation shift, including the CMNIST dataset, a trade-off exists between ID and OOD accuracy for all methods. However, as illustrated in Figure 10, for diversity shift, a positive correlation between ID and OOD accuracy was observed across all methods.

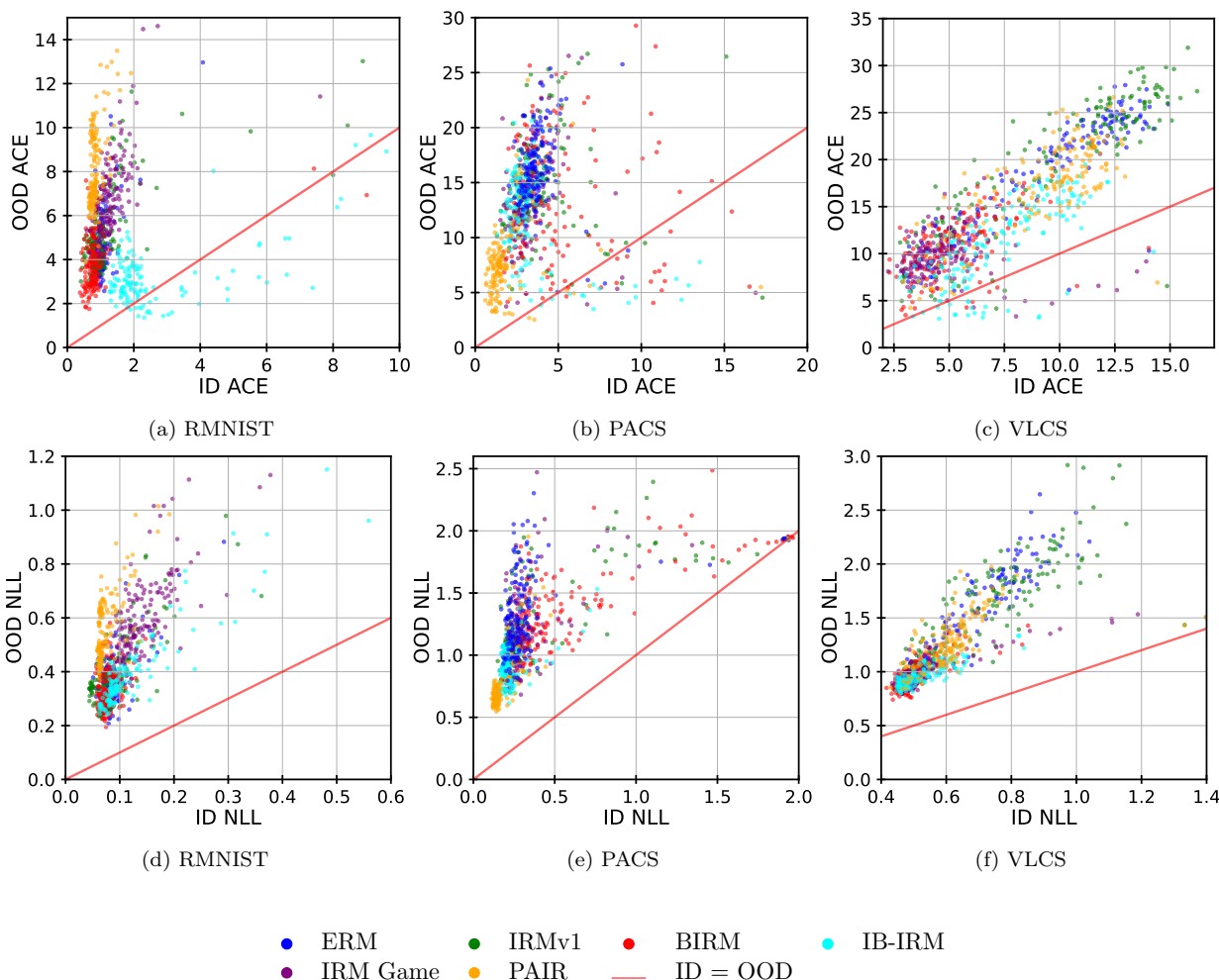

Figure 6: Comparison of the relationship between the calibration metrics in the training environment (horizontal axis) and the test environment (vertical axis). The red solid line represents the case where the calibration metrics is equal in both environments. It was observed that IB-IRM (in light blue) is distributed near the red solid line, indicating a tendency not to overfit to the training environment compared to other methods.

## C  Additional discussion and potential future directions

The experimental results demonstrated that IBIRM, which employs an information bottleneck, excels in terms of model calibration. This can be attributed to the fact that applying the information bottleneck restricts the representational power of the featurizer $\Phi(X)$, limiting its complexity and enabling it to discard spurious features, thereby reducing dependence on them. This approach of constraining the complexity of data representations has been applied in various contexts, and there are related studies in the field of Large Language Models (LLMs) as well as follows.

Ruan et al. (2021) demonstrated that applying the information bottleneck method to the vision-language model CLIP (Radford et al., 2021) improved its OOD generalization performance. Furthermore, Hu et al. (2021) proposed a new fine-tuning method called Low-Rank Adaptation (LoRA). This method involves attaching a very low-rank linear layer as an adapter to a pre-trained model and fine-tuning only that part for a specific task, achieving performance comparable to full fine-tuning with significantly lower memory requirements. This approach can be interpreted as compressing task-specific information into a low-rank representation during learning, suggesting a connection with the information bottleneck method.

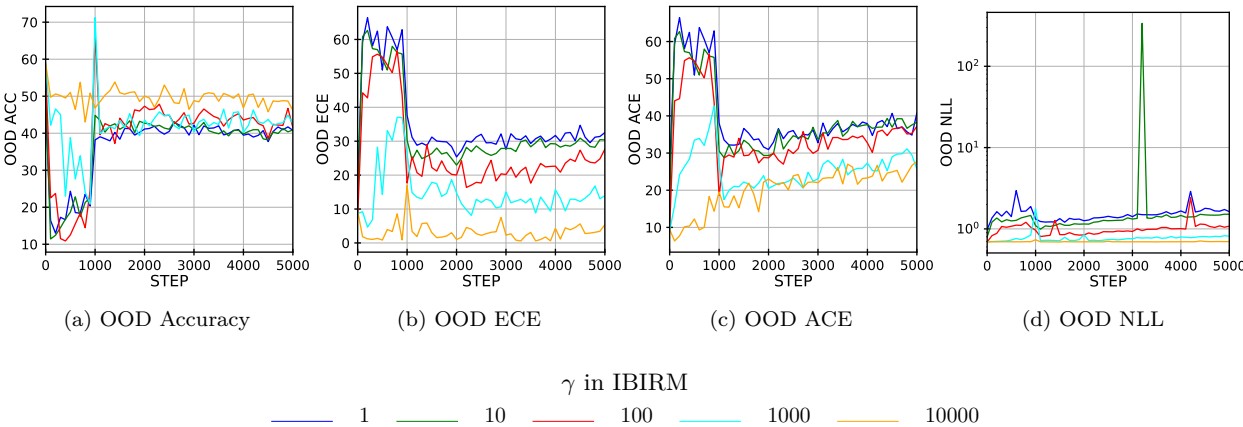

(a) OOD Accuracy     (b) OOD ECE     (c) OOD ACE     (d) OOD NLL

Figure 7: Comparison of the model's OOD calibration performance in CMNIST, with various values of $\gamma$ and $\lambda$ fixed at a sufficiently large value (10000) in the formulation of IBIRM 9. It was observed that as $\gamma$ increases, i.e., as the regularization penalty from the information bottleneck method becomes more significant, the calibration performance of the model in OOD scenarios improves.

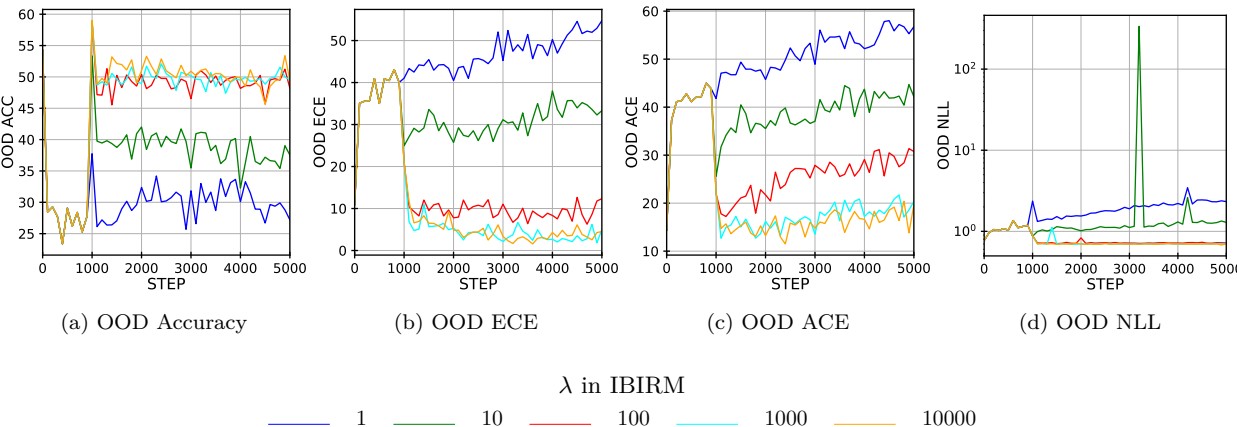

(a) OOD Accuracy     (b) OOD ECE     (c) OOD ACE     (d) OOD NLL

Figure 8: Comparison of the model's OOD calibration performance in CMNIST, with various values of $\lambda$ and $\gamma$ fixed at a sufficiently large value (10000) in the formulation of IBIRM 9. It was observed that as $\lambda$ increases, i.e., as the regularization penalty from IRMv1 in IB-IRM becomes more significant, the calibration performance of the model in OOD scenarios improves.

Based on the aforementioned research, future studies are anticipated to investigate the impact of learning through information compression on OOD generalization and calibration, as well as to develop novel methods that simultaneously address these two issues.

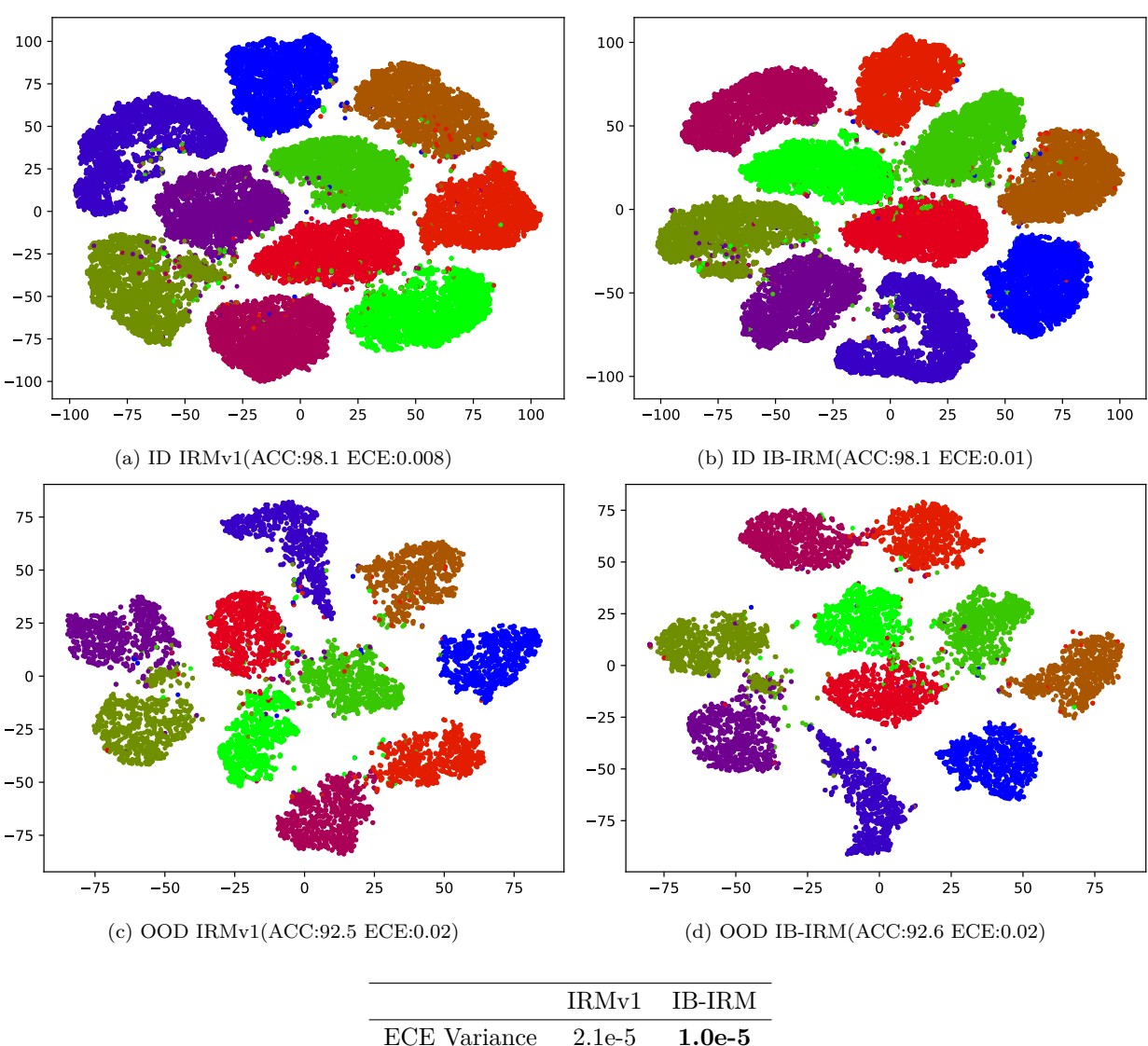

(a) ID IRMv1(ACC:98.1 ECE:0.008)

(b) ID IB-IRM(ACC:98.1 ECE:0.01)

(c) OOD IRMv1(ACC:92.5 ECE:0.02)

(d) OOD IB-IRM(ACC:92.6 ECE:0.02)

|  | IRMv1 | IB-IRM |
|---|---|---|
| ECE Variance | 2.1e-5 | **1.0e-5** |

Figure 9: Visualization of the latent feature spaces for IRMv1 and IB-IRM on RMNIST. The top row shows ID visualizations, and the bottom row shows OOD visualizations. The color of each point represents the class (RMNIST involves 10-class classification). When invariant features are properly acquired, the latent features should remain consistent across environments. IB-IRM, which achieves more stable ECE variance across environments, demonstrates stable feature distributions in both ID and OOD. In contrast, IRMv1, which exhibits worse ECE variance, shows variations in feature distributions between ID and OOD, suggesting it may be acquiring different features depending on the environment.

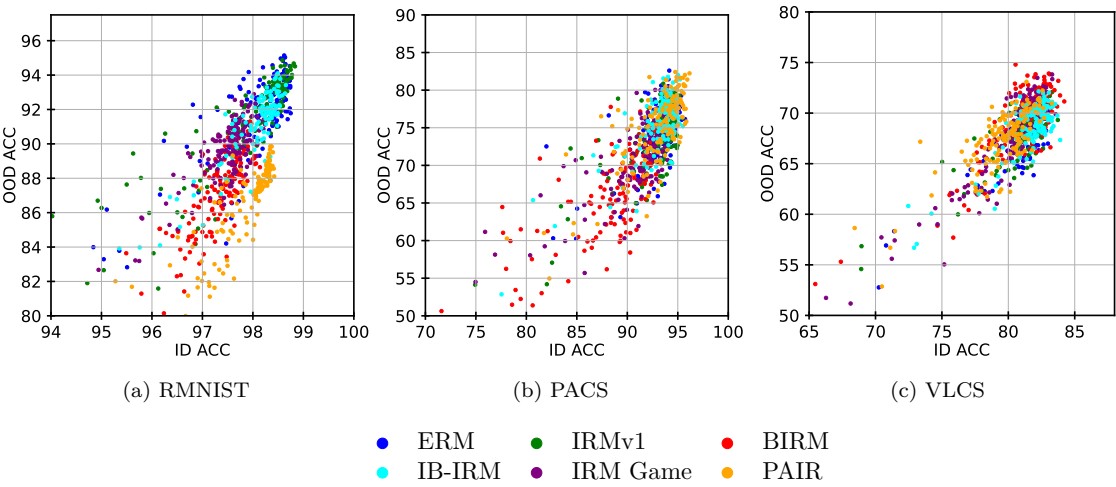

(a) RMNIST        (b) PACS        (c) VLCS

Figure 10: This figures illustrate the relationship between ID and OOD accuracy on each dataset of Diversity shift. It is evident that there exists a positive correlation between accuracy on the ID data and accuracy on the OOD data.

