# OpenReview forum: "Towards Understanding Variants of Invariant Risk Minimization through the Lens of Calibration"
_TMLR — Accepted by TMLR_

### Review · Reviewer_vARo · 2024-05-10

**Summary Of Contributions:**

The paper studies the interaction between confidence calibration and accuracy on out-of-distribution (OOD) data for a family of invariant risk minimization (IRM) methods. Specifically, it notices a trade-off between the calibration and accuracy of OOD data. The main takeaway is that accuracy alone cannot measure invariant feature learning. Future research should also take into account calibration.

**Audience:**

Yes

**Broader Impact Concerns:**

No major concerns.

**Claims And Evidence:**

Yes

**Requested Changes:**

* The main takeaway is unsatisfying in its current form. Specifically, it failed to convince me that calibration measures the quality of learned invariant features.

* It would be great if the authors could consider including other calibration metrics in the experiments.

**Strengths And Weaknesses:**

Strength:
* Out-of-distribution experiments are conducted on various distribution shifts.
* The paper carefully studies the effects of regularization on the accuracy and calibration of IRM methods on OOD data.

Weakness:
* **Confusing motivation**: The paper's main contribution is using the confidence calibration metric, expected calibration error (ECE), to measure the OOD generalization performance of invariant feature learning. It first establishes that good calibration is sufficient for OOD generalization by invoking the definition of calibration and OOD generalization. However, the experiments show that calibration does not always correlate with OOD data accuracy. Intrinsically, calibration is a measure of alignment between the uncertainty and predictions of a model. A model can be very robust on OOD data (high accuracy) while over-confident or under-confident (poor calibration). Does this mean this model is not robust? Conversely, a model can perform very poorly on OOD data (low accuracy) while being perfectly calibrated. As stated in the paper, good calibration is only a *sufficient* but not a *necessary* condition for good OOD generalization. This brings it to my question: why should we consider calibration as an important metric for invariant feature learning? since it cannot faithfully reflect a model's robustness and generalization on OOD data as the experiments suggested.

* **ECE is not the best metric for measuring confidence calibration**: Expected calibration error has many drawbacks. Several prior works focusing on measuring calibration have proposed improved metrics such as Adaptive calibration error (AECE) [1] and test-based calibration error (TCE) [2].

[1] Nixon, J., Dusenberry, M. W., Zhang, L., Jerfel, G., & Tran, D. (2019). Measuring Calibration in Deep Learning. In CVPR workshops (Vol. 2, No. 7).

[2] Matsubara, T., Tax, N., Mudd, R., & Guy, I. (2023). TCE: A Test-Based Approach to Measuring Calibration Error. In: Proceedings of the Conference on Uncertainty in Artificial Intelligence, PMLR

---

> ### Author Response · Authors · 2024-06-05
> **Response to reviewer vARO**
>
> Thank you for taking the time to review our manuscript and providing valuable feedback.
>
> > why should we consider calibration as an important metric for invariant feature learning?
>
> As you pointed out, calibration across environments is only a sufficient condition for model invariance. It is possible for a model to be invariant without being calibrated across environments.
>
> However, comparing the degree of calibration across environments is a necessary and sufficient measure of model invariance.
> The invariance condition can be restated as the model's actual prediction accuracy $E[Y_e|f(X_e)]$ is consistent across all environments, given its confidence level, meaning that the model's overconfident degree is consistent across all environments.
>
> Therefore, we can evaluate the invariance by evaluating the variation of model calibration across environments. We use ECE and ACE metrics as approximations of the model's calibration performance and evaluate the consistency of calibration performance across environments to compare and evaluate the invariance of IRM variants.
>
> Furthermore, our empirical results suggest that calibration performance is consistent across environments only when calibration performance is relatively better (See Fig2 and Fig4). This implies that calibration across environments is not only a sufficient condition for invariance but also a necessary condition empirically.
>
> We have revised the paper to ensure our research motivation is not confusing. Specifically, we have added detailed explanations in Section 2.3. We have also revised the experimental results. In Table 2, we previously compared variants based solely on the calibration performance in OOD. However, we now compare the variance of calibration performance across environments to evaluate the consistency of calibration performance. Even after the revision, IB-IRM consistently showed calibration across environments. Figure 2 has also been modified to compare calibration performance in ID and OOD.

---

> > ### Comment · Reviewer_vARo · 2024-06-15
> > **Re**
> >
> > Hi,
> >
> > Thank you for the response. However, I still have one question regarding using calibration to measure invariance. I have a hypothetical scenario in mind. Suppose a model is always over-confident by 0.1 in its prediction, i.e., $E[Y^e|f(X^e)=a] = min(1.0,a+0.1), \quad \forall e$, the model is robust and invariant according to eq.2 but poorly calibrated according to eq.3.
> >
> > Even though this might be a rare case, it is theoretically possible. How do you describe this model's invariance learning?

---

> > > ### Author Response · Authors · 2024-06-17
> > > **Response to reviewer vARO**
> > >
> > > Thank you for your valuable response.
> > >
> > > As you rightly pointed out, the scenario in which the degree of over-confidence is consistent across all environments, thereby satisfying the invariance of eq.2 but not the cross-environment calibration of eq.3, is theoretically possible. This is because eq.3 serves only as a sufficient condition for eq.2, not as a necessary one.
> > >
> > > However, as elucidated in Section 2.3, our focus is not on the degree to which eq.3 is satisfied, that is, the calibration quality of the model, but rather on the extent to which the calibration degree remains invariant across environments. Specifically, we assess the degree of invariant learning by examining whether the model exhibits a consistent level of over-confidence across all environments. In reference to your example, if a calibration metric indicates that the model is over-confident by 0.1 uniformly across all environments, it implies that the model satisfies eq.2, thus demonstrating invariance. We measured the variance in calibration across environments using the environment-wise dispersion of metrics such as ECE or ACE, to assess invariance (see Figures 2 and 4, and Table 2).
> > >
> > > Conversely, metrics that do not account for the relationship between model confidence and actual accuracy, such as accuracy or F1 score, fail to evaluate the model’s over-confidence degree and, consequently, cannot evaluate invariance in the sense of eq.2.

---

> > > > ### Comment · Reviewer_vARo · 2024-06-17
> > > > **Re**
> > > >
> > > > Hi,
> > > >
> > > > Thank you for the response. It makes sense to me now. It would be great if the paper provided some intuitive examples to help readers understand that **consistency and variance of calibration across environments measure invariance**.
> > > >
> > > > For example, this concept is not emphasized in the abstract and introduction. I was under the wrong impression that *better calibration indicates better invariance*, which is not true as we discussed.
> > > >
> > > > Thanks!

---

> > > > > ### Author Response · Authors · 2024-06-17
> > > > > **Response to reviewer vARO**
> > > > >
> > > > > Thank you for your suggestion.
> > > > >
> > > > > As you pointed out, it is important to make the explanation of the concepts we discussed more understandable. Therefore, we have revised the abstract and introduction in our paper accordingly, and we have also added the simple and clear example you provided to the end of the section 3.2, "Why do we use calibration to evaluate IRM variants?".
> > > > >
> > > > > We hope these second revisions address your concerns.

---

> ### Author Response · Authors · 2024-06-05
> **Response to reviewer vARO**
>
> > ECE is not the best metric for measuring confidence calibration
>
> Thank you for your suggestions. We have conducted additional evaluations using ACE and NLL metrics to reinforce our claim. The results were consistent with the evaluation using ECE alone. Notably, IB-IRM exhibited consistent calibration across environments in all metrics (See Figs. 2, 3, 4, 5, Table 4 and Appendix B, C).

---

### Review · Reviewer_GgdA · 2024-05-16

**Summary Of Contributions:**

This paper studies invariant risk minimization (IRM) by using Expected Calibration Error (ECE) as a matric to measure the learning of invariant features. The authors studies different variants of IRM and show which is observed to achieve both low ECE and high accuracy. Through empirical evaluation, the authors show the IB-IRM can have low ECE and maintaining original accuracy. Moreover, a tradeoff is identified to achieve robust generalization and meanwhile accurate calibration. The finding shows potential for the development of invariant learning by enhancing the prediction confidence without sacrificing accuracy.

**Audience:**

Yes

**Broader Impact Concerns:**

No ethical issues.

**Claims And Evidence:**

Yes

**Requested Changes:**

Please see weaknesses for detail.

**Strengths And Weaknesses:**

Strengths:
- This paper shows many intriguing findings for invariant learning, many IRM variants are extensively studied.
- The paper is well-organized.
- The empirical analysis is intuitive and inspiring.

Weaknesses:
- Lack of motivation during demonstrating the idea. Why using ECE as a measure and how can it act as an important measure for invariant learning performance should be more clearly explained. Moreover, why it is important to compare various IRM variants? The reason behind some methods achieving good ECE and accuracy results are not intuitively explained.

---

> ### Author Response · Authors · 2024-06-05
> **Response to reviewer GgdA**
>
> We appreciate your valuable feedback and your assessment of our work.
>
> > Why using ECE as a measure and how can it act as an important measure for invariant learning performance?
>
> Our primary goal in this study was to compare the invariance of different Invariant Risk Minimization (IRM) variants. We use the invariance definition from [1] as the consistency of the actual prediction accuracy $\mathbb{E}[Y_e|f(X_e)]$ across environments, given the model's confidence.
> This translates to comparing the degree of conditional probability across environments.
> Therefore, we evaluate invariance by assessing the consistency of calibration performance across environments. Precisely, we utilize calibration metrics such as ECE to measure the consistency of calibration performance across environments.
>
> Our empirical findings suggest that comparable calibration performance across environments holds only when the calibration performance is relatively better (See Fig 2, 4, and 6 ). This indicates that calibration consistency across environments is not only a sufficient condition for invariance but also appears to be a necessary condition empirically.
>
> Metrics like accuracy do not consider $\mathbb{E}[Y_e|f(X_e)$] and are insufficient for evaluating strict invariance conditions. In contrast, calibration metrics such as ECE  take $\mathbb{E}[Y_e|f(X_e)]$ into account, enabling us to evaluate invariance more rigorously.
>
> To address your concerns about the clarity of our research motivation, we have revised the manuscript to provide a more comprehensive explanation. Specifically, we have added detailed descriptions in Section 2.3. We have also revised the experimental results. Previously, Table 2 compared variants based solely on the quality of calibration performance in the OOD setting. However, we now compare the variance of calibration performance across environments to assess the consistency of calibration performance. Despite the changes, IB-IRM consistently exhibits robust calibration across environments. Figure 2 has also been revised to compare the relationship between calibration performance in the ID and OOD settings.
>
> We believe these modifications significantly enhance the clarity of our research motivation.
>
> [1] Martin Arjovsky, L’eon Bottou, Ishaan Gulrajani, and David Lopez-Paz. Invariant risk minimization. arXiv preprint arXiv:1907.02893, 2020.

---

> ### Author Response · Authors · 2024-06-05
> **Response to reviewer GgdA**
>
> > why is it important to compare various IRM variants? The reason behind some methods achieving good ECE and accuracy results are not intuitively explained.
>
> As mentioned in Section 3.2, IRM variants often struggle to outperform well-tuned ERM.
> It remains unclear to what extent they successfully learn invariant features and which approaches are preferable for invariant feature learning. By comparing invariance using calibration, we can evaluate invariance more rigorously than simply using accuracy. This contributes to defining the future research direction for IRM variants and invariant feature learning algorithms.
> We speculate that IBIRM's success in achieving calibration across environments stems from its use of entropy regularization during training, which reduces model complexity and prevents overfitting to the training domain. It is well known that model complexity can deteriorate calibration [2].
>
> In the CMNIST experiment, the superior calibration performance of IRMv1, IBIRM, and BIRM across environments can be attributed to their shared use of penalty-based approaches to prevent overfitting to specific environments. Regularization designed to prevent overfitting is known to improve calibration [2]
> We have added a discussion of these points in Section 6.
>
> We hope that these revisions address your concerns, and we kindly request you to reconsider our manuscript.
>
> [2] Guo, Chuan, et al. "On calibration of modern neural networks." International conference on machine learning. PMLR, 2017.

---

### Review · Reviewer_Loo1 · 2024-05-22

**Summary Of Contributions:**

This paper investigates variants of Invariant Risk Minimization (IRM) using Expected Calibration Error (ECE) as a key metric to measure how well the model learns domain-invariant features. The authors conduct comparative evaluations of these IRM variants on datasets with distributional shifts. They conclude that the degree of invariant feature learning cannot be fully captured by OOD accuracy alone, necessitating systematic generalization metrics. The paper also highlights the need to better understand the relationship between OOD generalization and uncertainty calibration to develop improved IRM variants.

**Audience:**

Yes

**Claims And Evidence:**

Yes

**Requested Changes:**

1. [Critical] Discuss additional evaluation metrics that could be used to assess the IRM variants more comprehensively. Consider metrics such as F1 score, AUC, Adaptive Calibration Error, Maximum Calibration Error, and Negative log-likelihood. To provide new insights, apply some of these metrics to at least one of the datasets used in the study. If certain metrics are deemed not useful for this particular investigation, discuss the reasons behind this decision.


2. [Suggested] Include feature visualizations to provide a deeper understanding of the learned features captured by the different IRM variants. One approach to consider is the visualization technique used in the paper "Enhancing Compositional Generalization via Compositional Feature Alignment" (https://arxiv.org/abs/2402.02851), which employs similar datasets for domain-invariant learning. By incorporating such visualizations, readers can gain insights into how these methods capture invariant features and how they differ from each other.

**Strengths And Weaknesses:**

## Strengths
The paper is an empirical investigation paper, so its value lies in its findings and observations. Here are some findings I think are interesting to the community of invariant feature learning:

1. Reveals that IRM variants with strong regularization typically achieve lower ECE.

2. Observes that Information Bottleneck-based IRM (IB-IRM) lowers ECE across environments, aligning more closely with IRM's original objectives while relatively maintaining accuracy. Discusses the theoretical basis for why the information bottleneck approach is effective for invariant feature learning.

3. Identifies a trade-off between ECE and accuracy for IRMv1, underscoring the compromises needed for robust generalization and calibration. Provides ablation studies that offer useful insights into the impact of different regularization terms in IB-IRM.

## Weaknesses
I think the major weakness is that the paper only uses ECE as a metric to evaluate variants of IRM, which makes the paper not deep enough. I understand this paper is an empirical investigation paper, and it indeed has some interesting findings. However, the paper could be strengthened by using more evaluation metrics other than accuracy and ECE to provide a more comprehensive picture, e.g., F1 score, AUC, Adaptive Calibration Error, Maximum Calibration Error, Negative log likelihood. Further, I think some feature visualization could also benefit the paper, as it could provide more insights into the learned features.

---

> ### Author Response · Authors · 2024-06-05
> **Response to reviewer Loo1**
>
> We thank the referee for their consideration and detailed review.
>
> >Additional experiments using more evaluation metrics
>
> Following your advice for obtaining more reliable results, we included evaluations using Adaptive Calibration Error (ACE) and Negative Log-Likelihood (NLL) in our paper. ACE is an improved calibration metric over ECE, while NLL indirectly observes calibration. Consequently, we obtained consistent results with those of ECE alone. Notably, IB-IRM achieved cross-environment calibration across all metrics.
> Regarding the F1 score you suggested, it primarily considers the ranking performance of the model. However, it does not fully capture the actual predictive accuracy given the model's confidence, $\mathbb{E}[Y_e | f(X_e)]$, which is crucial for evaluating invariance. Therefore, we decided not to adopt those metrics.
>
> We have added the above explanation at the beginning of Section 5 and included the results of the additional experiments(see Fig2, 5, 6, Tab2, and 4).
>
> We hope these revisions address your concerns and request that you reconsider our manuscript.

---

> ### Author Response · Authors · 2024-06-06
> **Response to reviewer Loo1**
>
> >some feature visualization could also benefit the paper
>
> Thank you very much for your valuable suggestion.
>
> We agree that feature visualization contributes to a clearer understanding of model invariance. Accordingly, we have undertaken a visualization study, which we present in Appendix B.3.
>
> We visualized the features learned by IRMv1 and IB-IRM on RMNIST. The visualization was performed by mapping the internal representations of the CNN to 2D using t-SNE. In an ideal scenario of invariant feature learning, the feature distributions should remain consistent across all environments.
>
> The results show that IB-IRM, which achieves more consistent calibration across all environments, demonstrates stable feature distributions in both ID and OOD settings. Conversely, IRMv1, which exhibits greater variability in calibration performance, shows changes in feature distributions between ID and OOD, indicating variability in the features extracted across different environments (see Fig 9).
>
> These findings suggest that consistency in calibration performance across environments is indeed reflective of the consistency in feature acquisition.

---

### Decision · Action_Editor_ooF9 · 2024-06-17

**Recommendation:** Accept as is

**Comment:**

This paper investigates the efficacy of approximate Invariant Risk Minimization techniques, as measured by calibration metrics measured across a variety of distribution shifts. The empirical results suggest that Information Bottleneck-based IRM is consistently well-calibrated, suggesting that it effectively captures environment-invariant features. The reviewers found the conclusions intriguing and paper and investigation well-organized. Some reviewers requested additional metrics, visualizations, and justifications, which were supplied in the author response and revision. The reviewers and I are now all in agreement that the paper meets the TMLR standards and should be published.

**Audience:**

Yes, some members of the community will be interested in this paper.

**Claims And Evidence:**

Yes, this paper's claims are supported by the evidence.